# Characteristics of the Rhizospheric AMF Community and Nutrient Contents of the Dominant Grasses in Four Microhabitats of the Subalpine Zone in Northwestern Yunnan, China

**Wei Li [1], Jiqing Yang [1], Fangdong Zhan [1], Jiawei Guo [1], Ya Zhang [2], Yong Ba [2], Hengwen Dong [3], Yongmei He [1,*]**

[1] College of Resources and Environment, Yunnan Agricultural University, Kunming 650201, China; ynauweili@126.com (W.L.); yjq.98@163.com (J.Y.); zfd97@ynau.edu.cn (F.Z.); guojia1579795@163.com (J.G.)

[2] Kunming General Survey of Natural Resources Center, China Geological Survey, Kunming 650100, China; zhangya@mail.cgs.gov.cn (Y.Z.); bayong@mail.cgs.gov.cn (Y.B.)

[3] Kunming Prospecting Design Institute of China Nonferrous Metals Industry Co., Ltd., Kunming 650051, China; donghengwen@zskk1953.com

[*] Correspondence: heyongmei06@126.com

**Abstract:** At the southeastern periphery of the Tibetan Plateau, the subalpine ecosystem hosts grasses as some of the most substantial species. However, the community and function of arbuscular mycorrhizal fungi (AMF) around the rhizospheres of grasses in the subalpine zone are still poorly understood. In the present study, 28 soils and 11 species of dominant grasses collected from four microhabitats (shrubland, grassland, woodland, and forest) in the subalpine zone of northwestern Yunnan, China, were used to investigate the AMF community by Illumina MiSeq high-throughput sequencing technology as well as nutrient contents. Among the four microhabitats, the maximum soil nutrient levels around the rhizospheres of grasses were observed in woodland. The nitrogen, phosphorus, and potassium concentrations in *Dactylis glomerata* shoots were significantly higher than those in the other 10 grass species. The AMF diversity of grassland in summer was substantially greater than that of the other three microhabitats ($p < 0.05$). Discrepancies were observed within a given plant species across microhabitats; for example, in summer, the nitrogen concentration in the shoot of *Iris tectorum* in woodland was significantly higher than that in both forest and shrubland ($p < 0.05$). A total of eight genera were detected in the AMF communities, which were dominated by *Glomus*, with a relative abundance of 45.4–94.4% in summer and 60.5–84.3% in winter. Moreover, the abundance of *Glomus* was significantly positively correlated with the content of alkali-hydrolyzable nitrogen in soil and nitrogen in grasses according to the Mantel test. As a critical nutrient element in soil, nitrogen is beneficial for plant growth. Thus, these results provide a better understanding of the resilience of soil AMF community and the ecological adaptability of grasses in the subalpine ecosystems of northwestern Yunnan.

**Keywords:** subalpine; arbuscular mycorrhizal fungi (AMF); community diversity; grasses; nutrient contents

## 1. Introduction

A crucial part of biodiversity preservation, climate regulation, water conservation, and other essential ecological services is performed within the subalpine ecosystem in worldwide terrestrial systems [1]. However, the subalpine ecosystem is confronted with a multitude of problems originating from both natural and anthropogenic sources. The fragility of the subalpine environment can be attributed to several factors, including the extensive arid hills and grassy slopes, the degradation of grassland, and the process of

desertification [2]. The coexistence of plants and microorganisms is a well-established phenomenon. Arbuscular mycorrhizal fungi (AMF) exhibit the potential to create mutually beneficial partnerships with approximately 80% of plant species and are commonly found in terrestrial ecosystems [3]. The presence of AMF has been observed to enhance plant development by means of interactions with the root system of the host plant [4]. In addition, AMF are a vital component for preserving ecosystem processes and have a widespread presence in many habitats, such as shrubland, woodland, and other ecosystems [5].

The subalpine region of northwestern Yunnan Province is dotted with large areas of grassland and woodland [6]. Shangri-La, situated on the southern periphery of the Qinghai–Tibet Plateau, is recognized as the central region encompassing the "Three Parallel Rivers," which is characterized by elevated terrain, frigid climatic conditions, and an intricate ecosystem [7]. In recent years, the alpine ecosystem of northwestern Yunnan has experienced serious damage due to the impact of human activities such as mineral mining [8], grazing [9], and tourism [10]. Previous studies have shown the significance of the AMF population in the process of vegetation restoration across subalpine ecosystems [11]. Qiang found that the AMF population is an instrumental contributor in shaping soil structure during the succession process in subalpine ecosystems [12]. Rodrigo suggested that AMF exhibit the potential to enhance the cultivation of plants and significantly contribute to the ecological restoration of tailings ponds [13]. However, few studies have been conducted on the structure and function of AMF communities in the subalpine ecosystems of northwestern Yunnan.

It is postulated that discernible variations exist in the community structure of AMF as well as the mineral nutrient composition across diverse subalpine herbaceous grasses. Additionally, a substantial correlation is anticipated to be present between those two variables. First, an assessment was conducted to ascertain the chemical properties of the surface soil across four distinct microhabitats. Next, we monitored the alterations in both the composition and functionality of AMF communities across several microhabitats. Finally, we identified links between soil properties and AMF community shifts. This study intends to explore the following aspects: (1) the characteristics of the mineral nutrients and rhizosphere soil chemical properties of dominant grasses in different microhabitats; and (2) the composition, diversity, and functional characteristics of the AMF communities around the rhizospheres of dominant grasses and their relationship to soil chemical properties and grass mineral nutrient contents.

## 2. Materials and Methods

### 2.1. Study Site

The experiment was conducted in Diqing Tibetan Autonomous Prefecture, Yunnan Province (26°52′–28°52′ N, 99°20′–100°19′ E). The total area is 11,613 km² with an average elevation of 3459 m (Figure 1). The site exhibits a cold temperate monsoon climate, characterized by a mean annual temperature of 6.9 °C, an extreme maximum temperature of 24 °C in June, and an extreme minimum temperature of −5.8 °C in January. The region consists of high altitude and low latitude with a mean annual precipitation of 589.3 mm [14].

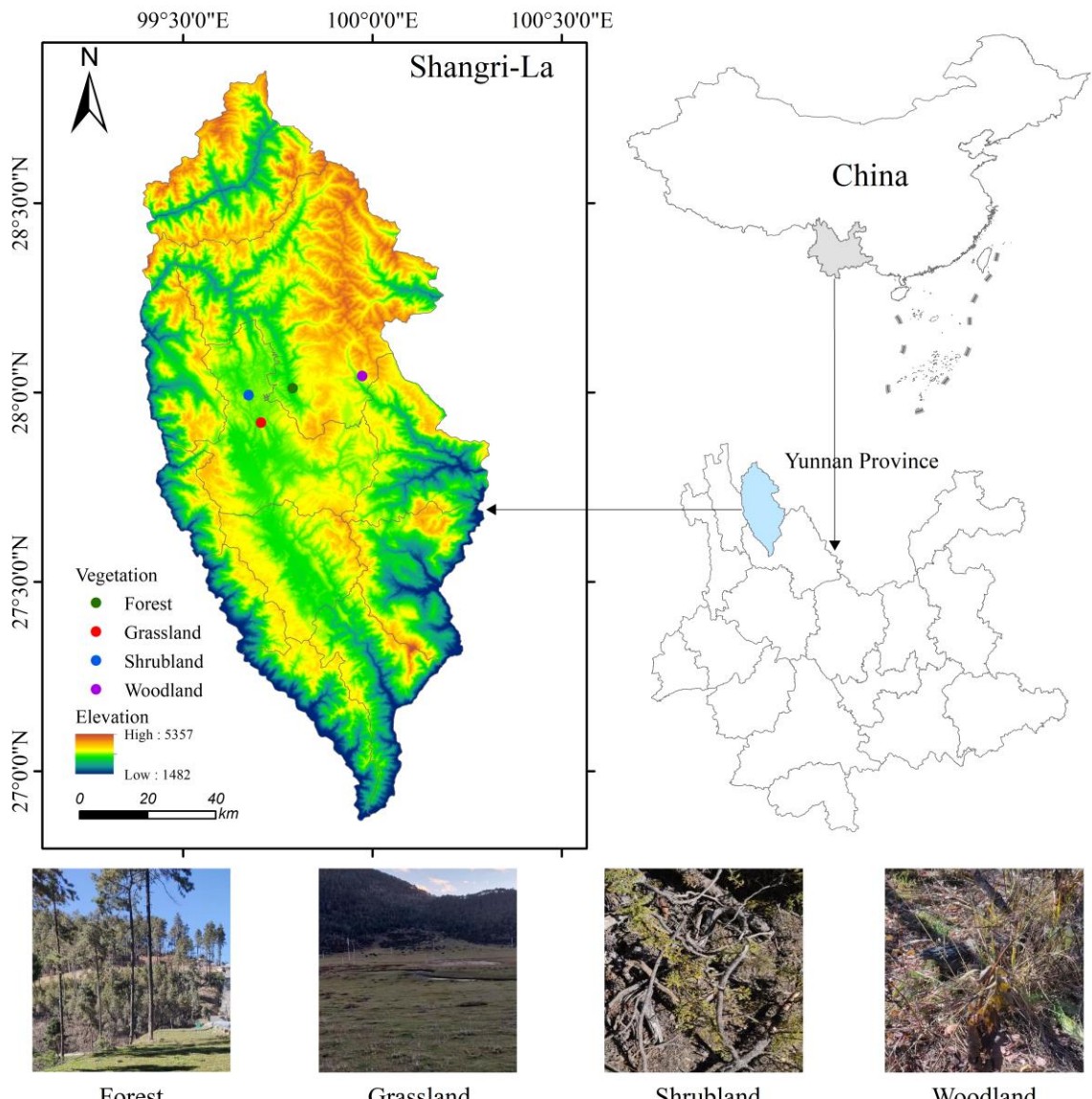

**Figure 1.** Locations of the 4 sites included in this study. Sites have been placed and marked by major vegetation types.

### 2.2. Sampling of Soil and Grasses

Grasses and their rhizosphere soil samples were randomly collected in four microhabitats, respectively, in November 2020 and June 2021. Among them, four grass species (*Stellera chamaejasme*, *Iris tectorum*, *Gentiana macrophylla*, and *Festuca pratensis*) were gathered in shrubland, four grass species (*Dactylis glomerata*, *Trifolium subterraneum*, *Poa annua*, and *Piptanthus concolor*) were gathered in grassland, two grass species (*Sambucus chinensis* and *Iris tectorum*) were gathered in woodland, and four grass species (*Sambucus chinensis*, *Iris tectorum*, *Gentiana macrophylla*, and *Primula forbesii*) were gathered in forest. Shrubland, grassland, woodland, and forest are both typical microhabitat types in subalpine areas of northwest Yunnan. Three experimental plots (1 × 1 m) were chosen in each microhabitat, and there was a minimum of 20 m between each plot. Soil samples from the root zone were taken at 5 different points in each experimental plot based on the shape of the plum blossom. First, grass roots were dug up from the ground, and the rhizosphere soil samples, which had a diameter of about 1 mm, were collected from the soil close to the roots, the 5 samples put on a plastic sheet, and mixed thoroughly. The soil drill was sanitized with

alcohol and cleaned with filter paper before and after each sampling to keep the samples from contaminating [15,16]. Three duplicate samples of rhizosphere soils were collected for each type of grass. Following removing of visible materials such as roots and litter, the rhizosphere soil samples were uniformly mixed after being sieved through a 2 mm screen. After being transferred to the lab, the rhizosphere soil samples were divided into 2 portions: one was promptly frozen at −80 °C for microbiological analysis, and the other was dried at room temperature for chemical properties test.

### 2.3. Sample Measurements

Grass samples were assessed separately by standard procedures. First, all samples were kept in paper bags, dried in an oven at 85 °C for 30 minutes, and then oven-dried at 70 °C to a constant weight. The samples were then crushed with a sample pulverizer, screened with a 40 mesh sieve, and mixed. Then, a 0.1 g (exact to 0.0001 g) sample was weighed into a 100 mL digestion tube, 1 mL of water was added for wetting, followed by addition of 5 mL of sulfuric acid into the digestion tube. The tube was shaken well, another 2 mL of hydrogen peroxide was added into the digestion tube twice, the tube was shaken well again, and then placed on the digestion furnace for digestion until the solution was colorless, which was then cooled to a constant volume of 100 mL. Nitrogen (N) was measured by the acid digestion-modified Kjeldahl method [17], phosphorus (P) was measured by the acid digestion-molybdenum antimony resistance colorimetric method [17], and potassium (K) was measured by the acid digestion-flame atomic absorption method [17].

For soil samples, soil organic matter (SOM) was measured by the potassium dichromate volumetric method [18]. Total nitrogen (TN) was measured by the acid digestion-indophenol blue colorimetric method [18], alkali-hydrolyzed nitrogen (AN) was measured by the alkaline hydrolysis diffusion method [18], total phosphorus (TP) was measured by the alkali fusion Mo-Sb anti-spectrophotometric method [18], available phosphorous (AP) was measured by the sodium hydrogen carbonate solution Mo-Sb anti-spectrophotometric method [18], and total potassium (TK) and available potassium (AK) were measured using the acid digestion-flame atomic absorption method [18].

### 2.4. Molecular Analysis of Soil Microorganisms

The diversity of the fungal community was determined and analyzed by Beijing Guoke Biotechnology Co., Ltd. by PCR amplification and high-throughput sequencing. The fungal internal transcribed spacer (ITS) region was amplified with the primers AMV4-5NF (5′-AAGCTCGTAGTTGAATTTCG-3′) and AMDGR (5′-CCCAAC-TATCCCTATTAATCAT-3′) (ABI, Gene®AMP9700, United States). After the genomic DNA was extracted by a soil kit (E.Z.N.A.®Soil DNA Kit, Omega Bio-tek, Norcross, GA, USA), it was detected by 1% agarose gel electrophoresis, and all samples were analyzed according to the formal experimental conditions, with three replicates per sample. All PCR reactions were carried out with 20 μL TransGen AP221-02: TransStart Fastpfu DNA Polymerase PCR System; 0.8 μL of forward primers (5 μM/μL) and reverse primers (5 μM/μL), respectively; and about 10 ng template DNA. Thermal cycling consisted of initial denaturation at 95 °C for 3 min, followed by 32 cycles of denaturation at 95 °C for 30 s, annealing at 55 °C for 30 s, and finally elongation at 72 °C for 45 s and 72 °C for 10 min, 10 °C until halted. After the PCR products of the same sample were mixed, they were detected by 2% agarose gel electrophoresis, recovered using an AxyPrepDNA gel recovery kit (AXYGEN, Corning, NY, USA), and eluted with Tris HCl. For 2% agarose electrophoresis detection, referring to the preliminary quantitative results of electrophoresis, the PCR products were quantitatively detected by a QuantiFluor™- ST blue fluorescence quantitative system (Promega, Madison, WI, USA).

*2.5. Data Analysis*

Differences in several factors, including soil chemical properties and grass nutrients, among the four microhabitats were determined by two-way analysis of variance (ANOVA) and Tukey's test. The above statistical analyses were performed by SPSS 25.0 (International Business Machines Corporation, Armonk, NY, USA).

Diversity metrics were calculated using the core-diversity plugin within QIIME2. Feature-level alpha diversity indices, such as observed operational taxonomic units (OTUs) and the Chao1, Shannon, and Simpson indices, were calculated to estimate the microbial diversity within an individual sample. The Mantel test was performed to reveal the association of AMF communities with environmental factors based on the relative abundances of microbial species at the genus level. Origin 2021 (OriginLab Corporation, Northampton, MA, USA) and OmicStudio (LC-Bio Technologies Co., Ltd., Hangzhou, China) were used for drawing.

## 3. Results

### 3.1. Nutrient Contents of Grass

*D. glomerata* in grassland had the largest shoot N content throughout the summer, at 37.80 g·kg$^{-1}$, far exceeding that of any other grass. Additionally, the mean concentration of N in the shoots of grass in grassland was comparatively greater than that observed in the other three microhabitats. Moreover, compared to other grasses, the P and K levels of *D. glomerata* were significantly higher (Figure 2).

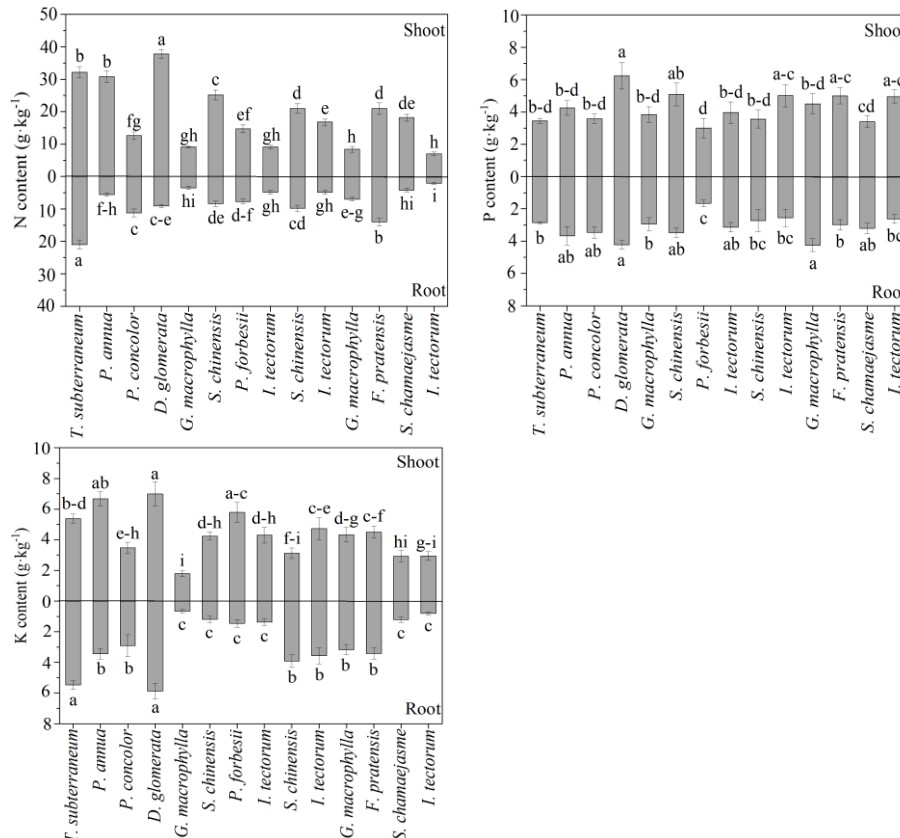

**Figure 2.** Nutrient contents of grasses in four microhabitats, measured in June 2021. N, nitrogen; P, phosphorus; K, potassium. Values followed by different letters (a–i) assigned to various bars along the same horizontal axis denote statistically significant differences in nutrient levels, with a significance level of $p < 0.05$ (Tukey test).

*T.* subterraneum was found in grassland and had the highest shoot N content during the winter season. Conversely, in shrubland, *S. chamaejasme* exhibited the highest root N

content. In particular, variations in N content were observed across different microhabitats. The P level of grass roots varied greatly among the four microhabitats, with shrubland possessing the lowest K (Figure 3).

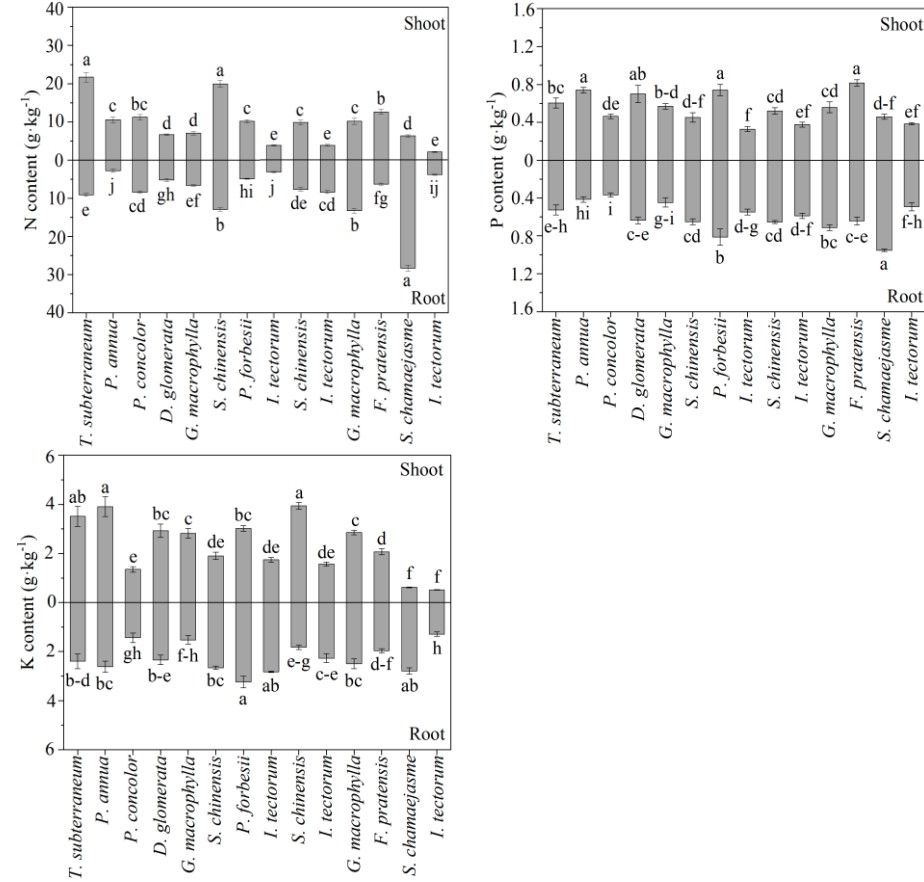

**Figure 3.** Nutrient contents of grasses in four microhabitats, measured in November 2020. N, nitrogen; P, phosphorus; K, potassium. Values followed by different letters (a–j) assigned to various bars along the same horizontal axis denote statistically significant differences in nutrient levels, with a significance level of $p < 0.05$ (Tukey test).

### 3.2. Soil Chemical Properties

The SOM in the four microhabitats ranged from 43.8 to 190.1 g·kg⁻¹, with the maximum content found in *I. tectorum* rhizosphere soil in woodland at 190.1 g·kg⁻¹. The TN content exhibited a similar trend, peaking at 6.4 g·kg⁻¹. The maximum TP and AP contents were found in the rhizosphere soil of *I. tectorum* in woodland, with 2.0 g·kg⁻¹ and 20.8 mg·kg⁻¹, respectively. Woodland soil had the largest nutrient content among the four microhabitats in summer (Table 1).

**Table 1.** Soil nutrients contents in summer.

| Habitat | Grass | SOM (g·kg⁻¹) | TN (g·kg⁻¹) | TP (g·kg⁻¹) | TK (g·kg⁻¹) | AN (mg·kg⁻¹) | AK (mg·kg⁻¹) | AP (mg·kg⁻¹) |
|---------|-------|--------------|-------------|-------------|-------------|--------------|--------------|--------------|
| Grassland | *T. subterraneum* | 43.8 ± 1.1 c | 2.1 ± 0.1 b | 0.8 ± 0.06 b | 18.5 ± 0.7 a | 98 ± 1.5 a | 174.5 ± 2.0 d | 4.2 ± 0.3 c |
| | *P. annua* | 62.5 ± 1.1 b | 2.8 ± 0.1 a | 0.9 ± 0.06 b | 10.1 ± 0.8 c | 84 ± 2.6 c | 205.0 ± 4.6 b | 9.6 ± 0.7 a |
| | *P. concolor* | 132.5 ± 3.4 a | 2.2 ± 0.1 b | 1.1 ± 0.10 a | 14.6 ± 1.0 b | 77 ± 2.4 d | 231.6 ± 3.1 a | 5.3 ± 0.3 b |
| | *D. glomerata* | 66.3 ± 1.9 b | 2.9 ± 0.1 a | 0.8 ± 0.06 b | 11.4 ± 1.2 c | 91 ± 2.3 b | 193.1 ± 4.4 c | 8.9 ± 0.6 a |
| Forest | *G. macrophylla* | 170.1 ± 3.7 b | 4.8 ± 0.3 c | 0.8 ± 0.10 b | 14.8 ± 0.5 b | 182 ± 3.2 c | 287.2 ± 3.1 b | 13.4 ± 0.3 a |
| | *S. chinensis* | 173.8 ± 2.7 b | 4.7 ± 0.3 c | 0.5 ± 0.06 c | 17.3 ± 0.6 a | 210 ± 3.8 b | 182.3 ± 2.3 d | 10.5 ± 1.0 b |
| | *P. forbesii* | 183.8 ± 3.4 a | 6.0 ± 0.3 a | 0.4 ± 0.0 d | 16.1 ± 1.1 ab | 280 ± 4.4 a | 224.1 ± 3.2 | 15.1 ± 1.3 a |

| | | | | | | | | |
|---|---|---|---|---|---|---|---|---|
| Woodland | *I. tectorum* | 176.3 ± 3.4 b | 5.5 ± 0.2 b | 0.9 ± 0.06 a | 15.3 ± 0.7 b | 217 ± 3.2 b | 298.0 ± 4.1 a | 14.2 ± 0.6 a |
| | *S. chinensis* | 187.6 ± 5.0 a | 6.2 ± 0.3 a | 2.0 ± 0.10 a | 14.8 ± 0.7 a | 224 ± 3.9 a | 181.2 ± 2.1 a | 20.8 ± 1.0 a |
| | *I. tectorum* | 190.1 ± 4.5 a | 6.4 ± 0.3 a | 0.7 ± 0.06 a | 16.1 ± 0.8 a | 210 ± 3.9 a | 173.7 ± 2.7 a | 19.4 ± 0.9 a |
| Shrubland | *G. macrophylla* | 127.5 ± 2.7 a | 5.6 ± 0.3 a | 0.8 ± 0.06 a | 16.6 ± 0.6 b | 175 ± 4.3 a | 234.3 ± 3.9 a | 7.3 ± 0.3 b |
| | *F. pratensis* | 111.3 ± 3.4 b | 3.2 ± 0.2 c | 0.6 ± 0.10 ab | 11.7 ± 0.8 c | 182 ± 4.2 a | 149.8 ± 3.6 c | 6.1 ± 0.3 c |
| | *S. chamaejasme* | 116.3 ± 2.0 b | 4.3 ± 0.3 b | 0.6 ± 0.06 b | 18.5 ± 0.9 a | 147 ± 2.4 b | 236.0 ± 1.3 a | 9.1 ± 0.3 a |
| | *I. tectorum* | 66.3 ± 2.6 c | 2.7 ± 0.2 d | 0.7 ± 0.00 ab | 11.9 ± 0.5 c | 119 ± 3.5 c | 167.0 ± 3.0 b | 8.6 ± 0.6 a |

SOM, soil organic matter; TN, total nitrogen; TP, total phosphorus; TK, total potassium; AN, alkali-hydrolyzed nitrogen; AK, available potassium; AP, available phosphorus. Values presented in Table refer to the "mean ± standard deviation" (*n* = 3); values followed by different letters (a–d) in the same row are significantly different among microhabitats with *p* < 0.05 (Tukey test).

The soil nutrient content values around the roots of the dominant grasses in the four microhabitats in winter followed the same pattern as that in the summer, all of which were highest in woodland. The SOM content in woodland averaged 143.6 g·kg⁻¹, which was higher than that in forest, shrubland, and grassland. The contents of TN, AN, AK, and AP were the highest around the rhizosphere soil of *S. chinensis* in woodland, which were 7.7 g·kg⁻¹, 794.5 mg·kg⁻¹, 57.9 mg·kg⁻¹, and 28.8 mg·kg⁻¹, respectively (Table 2).

**Table 2.** Soil nutrients contents in winter.

| Habitat | Grass | SOM (g·kg⁻¹) | TN (g·kg⁻¹) | TP (g·kg⁻¹) | TK (g·kg⁻¹) | AN (mg·kg⁻¹) | AK (mg·kg⁻¹) | AP (mg·kg⁻¹) |
|---|---|---|---|---|---|---|---|---|
| Grassland | *T. subterraneum* | 37.5 ± 0.8 a | 2.4 ± 0.2 a | 0.9 ± 0.06 a | 13.3 ± 1.1 a | 269.5 ± 10.5 a | 20.4 ± 0.9 b | 5.8 ± 0.6 c |
| | *P. annua* | 14.4 ± 1.2 c | 1.4 ± 0.1 b | 0.7 ± 0.06 b | 12.9 ± 0.6 a | 119.0 ± 3.5 b | 16.3 ± 1.1 c | 11.7 ± 0.7 b |
| | *P. concolor* | 21.9 ± 0.6 b | 0.9 ± 0.1 c | 0.4 ± 0.00 c | 12.9 ± 0.5 a | 106.7 ± 1.8 c | 20.0 ± 1.8 b | 4.1 ± 0.3 d |
| | *D. glomerata* | 15.6 ± 0.9 c | 1.5 ± 0.1 b | 0.8 ± 0.06 a | 13.9 ± 0.6 a | 124.2 ± 1.8 b | 43.4 ± 1.0 a | 14.1 ± 0.6 a |
| Forest | *G. macrophylla* | 64.5 ± 1.9 d | 3.8 ± 0.2 d | 0.9 ± 0.20 a | 13.4 ± 0.9 a | 423.5 ± 3.5 c | 22.2 ± 1.8 c | 10.6 ± 0.6 c |
| | *S. chinensis* | 85.8 ± 2.6 c | 4.7 ± 0.2 c | 1.1 ± 0.10 a | 11.8 ± 0.5 b | 465.5 ± 6.0 b | 31.2 ± 2.2 b | 11.4 ± 0.9 c |
| | *P. forbesii* | 187.5 ± 3.4 a | 6.3 ± 0.3 a | 1.0 ± 0.06 a | 11.6 ± 0.7 b | 526.7 ± 4.1 a | 34.8 ± 0.4 a | 16.5 ± 0.8 b |
| | *I. tectorum* | 167.6 ± 3.0 b | 5.2 ± 0.2 b | 0.9 ± 0.06 a | 7.9 ± 0.9 c | 535.5 ± 5.6 a | 33.8 ± 1.4 ab | 24.5 ± 1.1 a |
| Woodland | *S. chinensis* | 163.6 ± 3.2 a | 7.7 ± 0.4 a | 0.8 ± 0.06 a | 13.1 ± 0.6 a | 794.5 ± 4.1 a | 57.9 ± 2.1 a | 28.8 ± 1.0 a |
| | *I. tectorum* | 123.7 ± 2.8 a | 5.9 ± 0.3 a | 1.1 ± 0.06 a | 11.5 ± 1.0 a | 509.2 ± 8.7 a | 37.3 ± 1.4 a | 13.4 ± 2.0 a |
| Shrubland | *G. macrophylla* | 28.3 ± 1.3 c | 3.7 ± 0.2 a | 0.8 ± 0.06 a | 11.3 ± 0.9 b | 260.7 ± 8.4 b | 17.8 ± 0.6 a | 5.1 ± 0.4 d |
| | *F. pratensis* | 30.3 ± 1.1 bc | 1.2 ± 0.1 d | 0.8 ± 0.06 a | 13.0 ± 0.7 ab | 236.2 ± 5.4 c | 10.6 ± 0.8 c | 12.0 ± 0.5 a |
| | *S. chamaejasme* | 32.7 ± 1.6 b | 2.0 ± 0.1 c | 0.7 ± 0.00 b | 13.3 ± 1.4 ab | 215.2 ± 3.3 d | 9.1 ± 0.4 d | 8.6 ± 0.3 c |
| | *I. tectorum* | 37.9 ± 0.7 a | 2.8 ± 0.2 b | 0.9 ± 0.06 a | 13.5 ± 1.0 a | 281.7 ± 3.7 a | 12.5 ± 0.8 b | 10.3 ± 0.6 b |

SOM, soil organic matter; TN, total nitrogen; TP, total phosphorus; TK, total potassium; AN, alkali-hydrolyzed nitrogen; AK, available potassium; AP, available phosphorus. Values presented in Table refer to the "mean ± standard deviation" (*n* = 3); values followed by different letters (a–d) in the same row are significantly different among microhabitats with *p* < 0.05 (Tukey test).

### 3.3. Alpha Diversity of AMF Communities

The ACE index and Chao1 index exhibited their highest values in shrubland in summer. Additionally, the rhizosphere soil of the shrubland grasses exhibited the maximum AMF community abundance. The maximum Shannon index value was found in shrubland, suggesting that the rhizosphere soil of the grass species in shrubland had the greatest AMF community diversity (Figure 4). Additionally, a declining trend in soil AMF community diversity was observed across the habitat types in the following order: shrubland > woodland > grassland > forest.

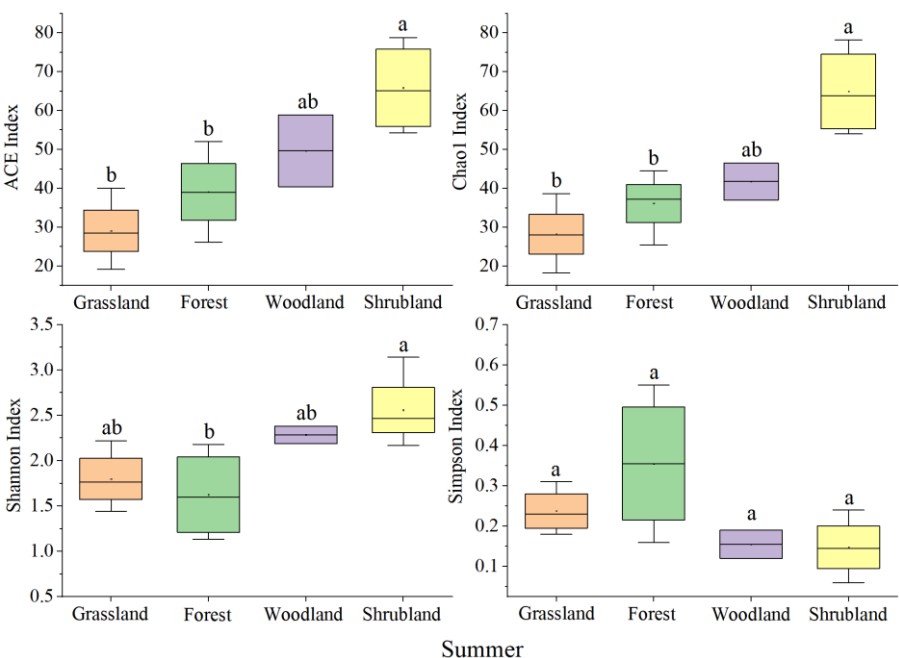

**Figure 4.** Analysis of soil AMF community diversity and richness in different habitats in summer. The transverse line in the box plot represents the mean; values followed by different letters (a,b) assigned to various bars along the same horizontal axis denote statistically significant differences in nutrient levels, with a significance level of $p < 0.05$ (Tukey test).

Although the soil AMF community was most abundant in the rhizosphere soil of the main grass in woodland throughout the winter, there were no significant differences in abundance or diversity among the four microhabitats, differing from the findings for summer (Figure 5).

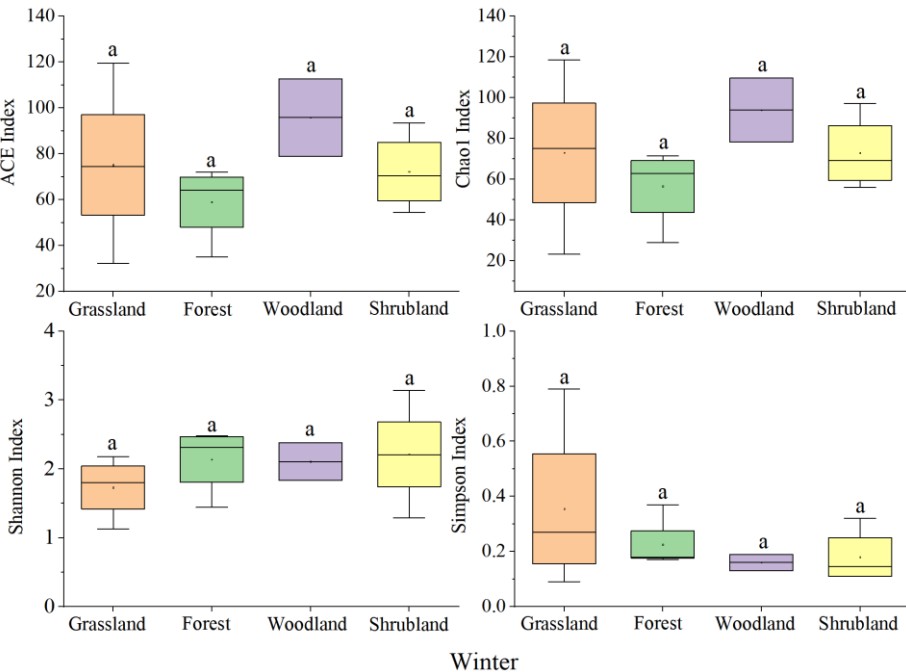

**Figure 5.** Analysis of soil AMF community diversity and richness in different habitats in winter. The transverse line in the box plot represents the mean; values followed by different letters (a,b) assigned to various bars along the same horizontal axis denote statistically significant differences in nutrient levels, with a significance level of $p < 0.05$ (Tukey test).

*3.4. Beta Diversity of AMF Communities*

A total of 188 classifiable OTUs were obtained by high-throughput sequencing technology in summer, of which 84 were from forest, 60 from woodland, 65 from grassland, and 126 from shrubland. The total number of OTUs shared among the four microhabitats was 16, which represents approximately 8.5% of the overall OTU count.

In comparison to the summer season, a total of 288 were found in the winter season, representing a 1.5-fold increase in the number of OTUs. Specifically, there were 16 OTUs unique to forests, representing 5.6% of the total, 22 OTUs unique to woodlands, accounting for 7.6%, 24 OTUs unique to grasslands, accounting for 8.3%, and 38 OTUs unique to shrublands, accounting for 13.2% of the total (Figure 6).

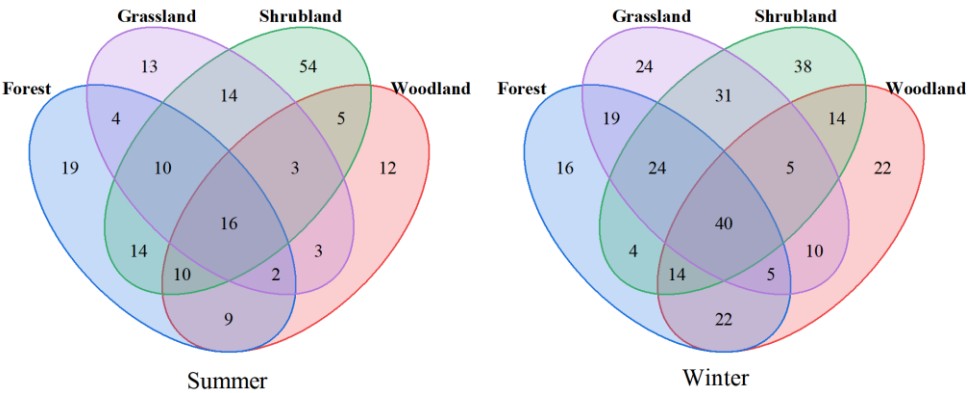

**Figure 6.** Analysis of β diversity of soil AMF communities in summer and winter.

*3.5. Composition of AMF Communities*

Fungi occurring during the summer season belonged exclusively to the *Glomeromycota*. The rhizosphere soil of the dominant grass in the four microhabitats contained an AMF community consisting primarily of three genera: *Glomus*, *Diversispora*, and *Paraglomus*. Moreover, woodland, shrubland, grassland, and forest were dominated by *Glomus*, accounting for 59.3%, 45.4%, 94.4%, and 46.4% (Figure 7), respectively.

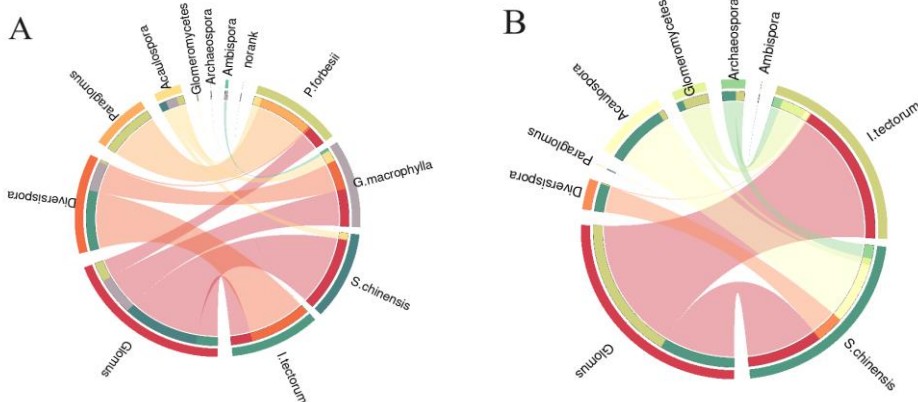

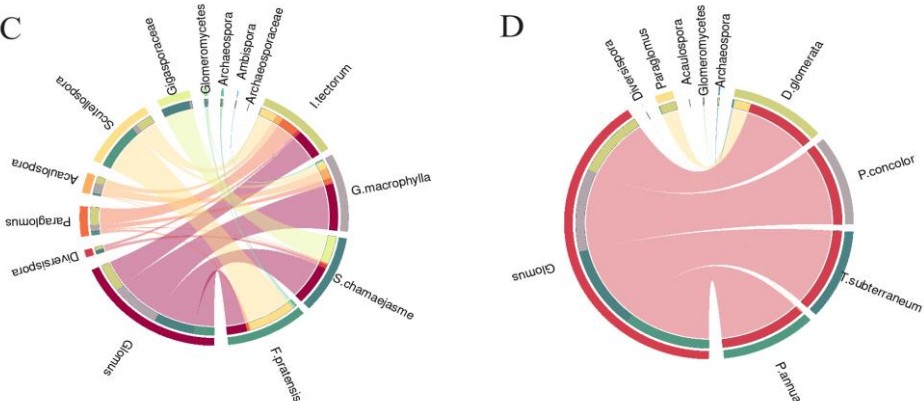

**Figure 7.** Taxonomic composition and distribution of AMF communities: the AMF communities distribution of the study area in summer, forest (**A**), woodland (**B**), shrubland (**C**), and grassland (**D**). *Glomus* had the richest abundance among the four microhabitats in summer.

A similar pattern to that discovered in summer was identified at the phylum level in winter. The three most prevalent genera in the AMF community were *Glomus*, *Acaulospora*, and *Paraglomus*. In accordance with the summer season, *Glomus* displayed the highest proportion, with percentages of 73.0%, 67.0%, 84.3%, and 60.5% in woodland, shrubland, grassland, and forest habitats (Figure 8), respectively.

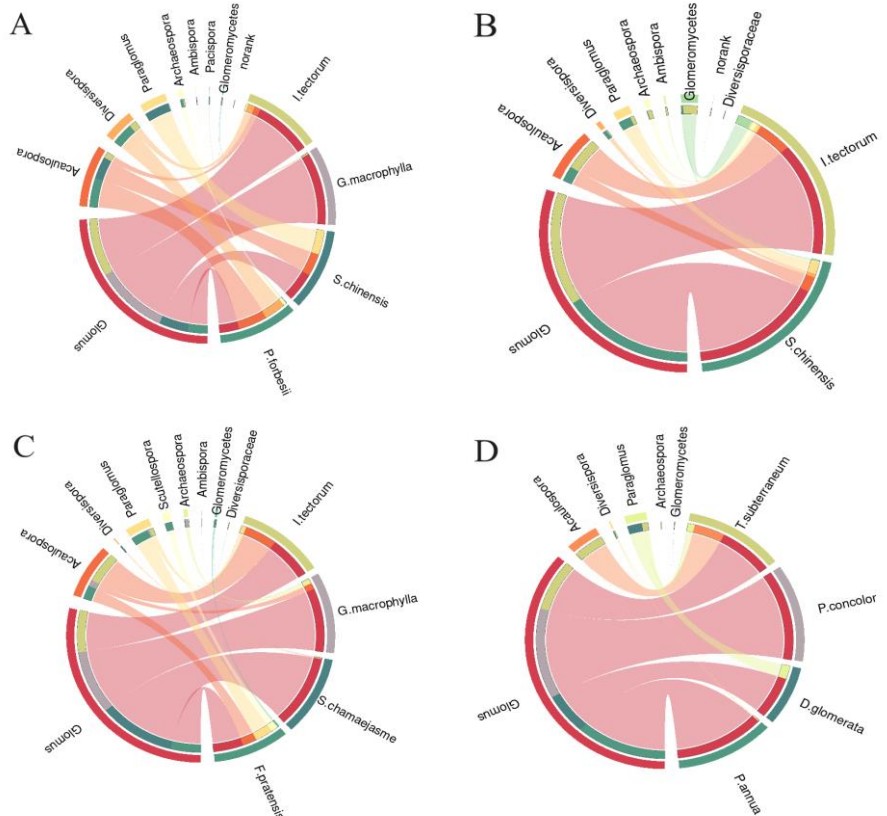

**Figure 8.** Taxonomic composition and distribution of AMF communities: the soil AMF communities distribution of the study area in winter, forest (**A**), woodland (**B**), shrubland (**C**), and grassland (**D**). *Glomus* had the richest abundance among the four microhabitats in winter.

### 3.6. Correlation Analysis of Soil Chemical Properties and AMF Communities

The Mantel test was conducted to examine the relationship between soil chemical properties and the dominant fungal groups at the genus level. The results revealed that there was a significant positive correlation between *Glomus* and *Paraglomus* and AN. Additionally, *Paraglomus* showed a negative correlation with AK. Moreover, *Diversispora* displayed a significant positive correlation with AP but a negative correlation with TK (Figure 9A).

Notably, only *Glomus* presented a positive correlation with soil TP during the winter period. However, *Glomus* presented a negative correlation with AK and AP. Similar trends also emerged for *Acaulospora* and *Paraglomus* (Figure 9B). This finding was associated with an increase in apoplastic substances during the winter season, as well as diminished growth rates of plants resulting from lower temperature and precipitation relative to the summer season [19].

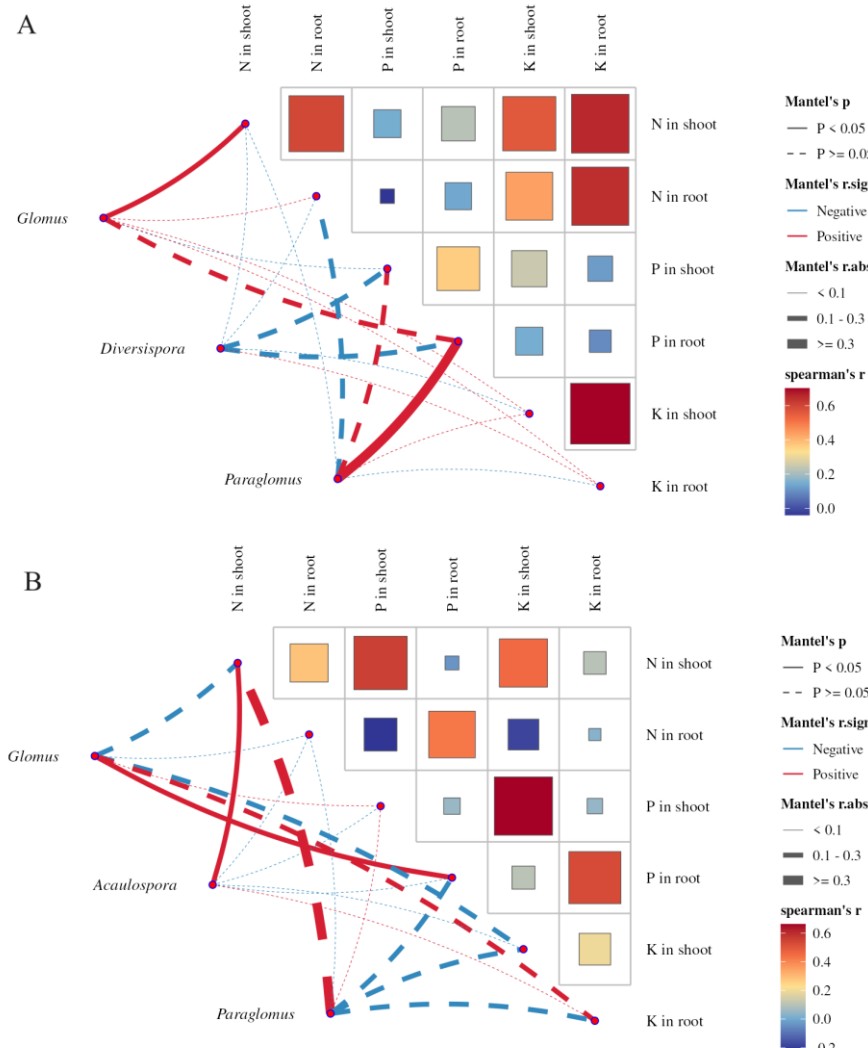

**Figure 9.** N, nitrogen; P, phosphorus; K, potassium. Mantel test of dominant AMF communities with soil chemical properties in summer (**A**); *Glomus* was significantly correlated with soil AN (Mantel test: r = 0.33, *p* < 0.05); *Paraglomus* was significantly correlated with soil AN (Mantel test: r = 0.37, *p* < 0.05); *Diversispora* was significantly correlated with soil AP (Mantel test: r = 0.37, *p* < 0.05). Mantel test of dominant AMF communities with soil chemical properties in winter (**B**); *Glomus* was significantly correlated with soil TP (Mantel test: r = 0.28, *p* < 0.05).

### 3.7. Correlation Analysis of Grass Nutrients and AMF Communities

*Glomus*, the dominant fungus, was positively correlated with P and K in whole grass and highly positively correlated with N in shoots in summer (Figure 10A). Thus, *Glomus*, the dominant fungi in the rhizosphere soil in the subalpine zone of northwestern Yunnan, had a positive influence on the accumulation of mineral nutrients (nitrogen, phosphorus, etc.).

*Glomus* and *Acaulospora*, the dominant fungi, demonstrated a positive correlation with mineral elements in grass in the winter, specifically N and P. This correlation exhibited a similar pattern to that observed in the summer.

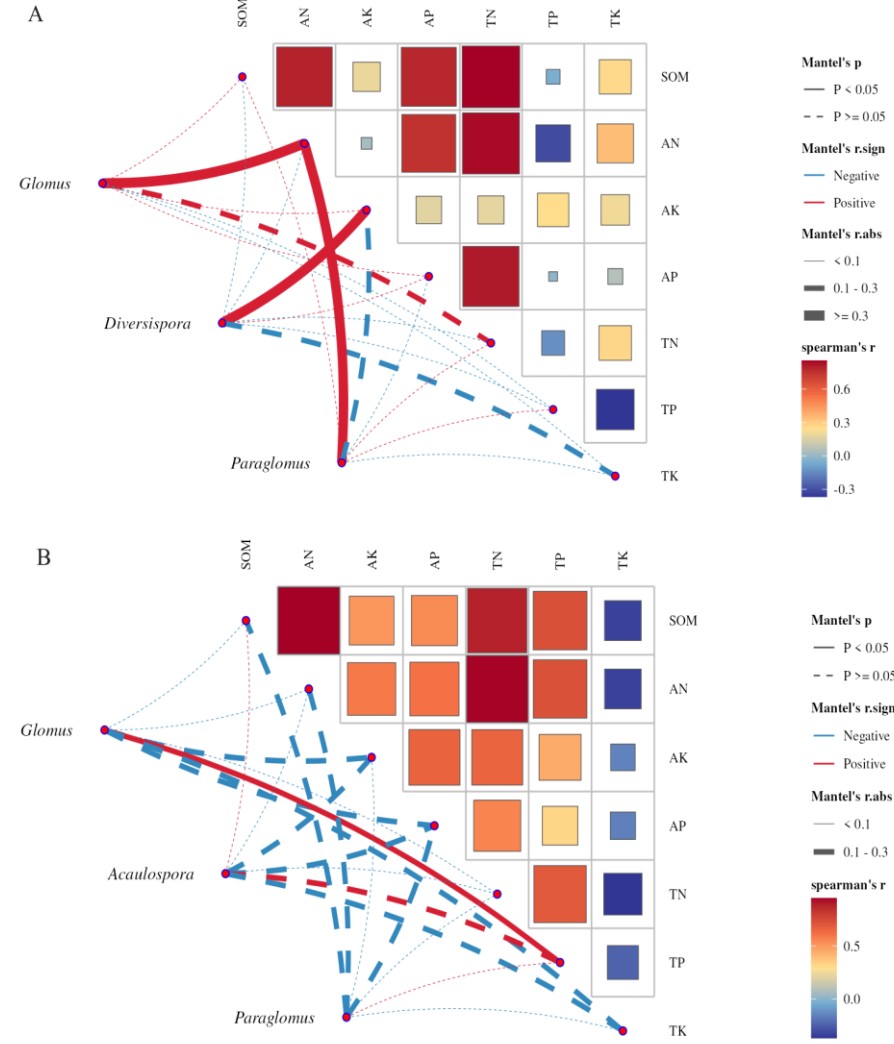

**Figure 10.** SOM, soil organic matter; TN, total nitrogen; TP, total phosphorus; TK, total potassium; AN, alkali-hydrolyzed nitrogen; AK, available potassium; AP, available phosphorus. Mantel test of dominant AMF communities with grass nutrients in summer (**A**); *Glomus* was significantly correlated with N in shoot (Mantel test: r = 0.17, $p < 0.05$); *Paraglomus* was significantly correlated with P in root (Mantel test: r = 0.60, $p < 0.05$). Mantel test of dominant AMF communities with grass nutrients in winter (**B**); *Glomus* was significantly correlated with P in root (Mantel test: r = 0.27, $p < 0.05$); *Acaulospora* was significantly correlated with P in shoot (Mantel test: r = 0.23, $p < 0.05$).

## 4. Discussion

### 4.1. Characteristics of Soil Chemical Properties and Grass Nutrients

Soil serves as the fundamental substrate for the sustenance of terrestrial plants and plays a crucial role in facilitating the exchange of matter and energy within ecosystems [20]. Soil microorganisms serve as significant indicators of soil fertility and quality [21], and they play a pivotal role in enhancing soil fertility and maintaining ecosystem stability

[22]. Altering the soil chemical properties and increasing the diversity of the soil microbiological community are beneficial to improving soil fertility. The results of this study show that soil nutrients were higher in summer than in winter. The woodland microhabitat exhibited the highest soil nutrient content among the four microhabitats. This can be attributed to the underground mining method employed at the Pulang Copper Mine, which minimizes surface soil disturbance in the mining area. In contrast, the forest microhabitat, being part of a forested area, contained higher levels of humus and experienced less external interference. The soil nutrient levels of shrubland and grassland were found to be low. This can be attributed to the fact that the grassland was classified as a human-made restoration zone following highway construction, characterized by significant human influence and limited vegetation cover.

Plant development is contingent upon the nutrients in the soil, and the health of plants is intimately associated with rhizosphere microorganisms. The rhizosphere microbial population is reliant upon the presence of plant roots as it is through this interaction that rhizosphere microorganisms and plants cohabitate and engage with one another [23]. The structure of the rhizosphere microbial community has a direct impact on plant health as it influences the uptake of nutrients inside the rhizosphere. In addition, the rhizosphere microbial community structure is affected not only by plants but also by soil types, altitude, plant habitat, and other factors [24]. Changes in soil nutrients, which are influenced by plant growth, impact the growth and development of plants [25].

In contrast to the summer season, the nutrient content of grass during winter exhibited a general decline. The TN content in the roots of woodland and shrubland grasses was higher than that in the shoots in winter. Additionally, the TP and TK contents in the roots of woodland, shrubland, and forest grasses were higher than those in the shoots. This difference was attributed to the thriving growth of grass during the summer season, as well as the slow growth and accumulation of litter in winter.

*4.2. Response of AMF Communities to Seasonal Changes*

The composition of the AMF community is influenced by the types of plants present [26] but also by climate factors [27]. The observed fluctuations in soil temperature and humidity are directly attributed to seasonal variations. D'Souza found that the AMF community composition, spore density, abundance, and diversity index were the largest in the premonsoon season and smallest in the monsoon season, and the AMF community and plants interacted with each other [28]. The composition of the soil fungal communities in the rubber forest studies underwent alterations as a result of seasonal fluctuations. The $\alpha$ diversity of AMF exhibited a significant increase during the rainy season as opposed to the dry season. This observed difference in diversity can be attributed to the larger amount of rainfall experienced during the rainy season, which created a more favorable habitat for microorganisms [29]. One factor contributing to the alteration in fungal community structure is depth; as the soil depth changes, so do soil nutrient levels, causing alterations to the soil fungal community [30]. The study of Bainard revealed that differences in the AMF community were caused by environmental parameters such as soil phosphate concentration, water content, and the type of host plant [31]. Significant variations were observed in the total infection rate, clump infection rate, vesicular infection rate, and soil spore density of the root system of *Caragana korshinskii* across different seasons. Furthermore, the seasonal differences in the community composition of AMF that surround the rhizosphere system were predominantly driven by soil TP, AP, and organic carbon levels [32]. However, Mónica [33] noted that the host plant phenology is the primary factor influencing seasonal variations in AMF spore density and spore community structure.

The present study observed higher values for the ACE index and Chao1 index during the winter than the summer. Additionally, the abundance of the AMF community was found to be greater during winter as opposed to summer, which aligns with the findings reported by Pang [34] in their previous study but contrasts with the results of the study by Wei [29]. The results can be attributed, in part, to the location of the study site within

the subalpine region, which is characterized by a diverse range of climate types. Additionally, the potential influence of several unquantified environmental variables, including predation, competition, and reciprocal symbiosis among microbial species [35,36], as well as ecological mechanisms involving dormancy and persistence traits shown by microbial communities and their constituents, would have contributed to the findings [37]. Long-term monitoring of microbial communities is needed to understand the seasonal dynamics and cycle patterns of these microorganisms as their distribution is generally influenced by a variety of environmental, vegetation, and other factors.

### 4.3. Effects of AMF Communities on Soil Chemical Properties

AMF are an essential component of the ecosystem and exert a substantial influence on the fertility and quality of soil [38]. AMF play a crucial role in the preservation of natural ecosystems by facilitating the maintenance of their functionality and stability. These fungi serve as conduits for the absorption of essential resources from the soil, including water, nitrogen, and phosphorus, which are subsequently made available to plants. In return, plants supply photosynthetic products to sustain the growth and reproductive processes of AMF [39].

In the present study, *Glomus* and *Paraglomus*, the dominant AMF in northwestern Yunnan, were significantly positively correlated with the content of AN, *Diversispora* was significantly positively correlated with the content of AP in summer, and *Glomus* was significantly positively correlated with the content of TP in winter. *Glomus* has been identified as the dominant genus in various unique environments, including mining sites, desert and dry habitats, and saline habitats. It exhibits remarkable adaptability and multiple functional attributes, making it an effective strain capable of withstanding stressful conditions [40]. In addition to inhibiting potato blight damage, *Glomus* stimulates potato development and elicits defense responses [41]. Moreover, *Glomus* has been shown to significantly reduce the harm caused to cucumber roots by root-knot nematodes. *Glomus* exhibited a 41% inhibitory effect on root-knot nematode injury to cucumber roots [42]. *Paraglomus* was found in multiple crop and grassland habitats with a wide range of land use intensities, soil pH values, soil textures, and phosphorus availability [43].

### 4.4. Effects of AMF Communities on Grass Nutrients

AMF invade and colonize plant roots, establishing a mutualistic symbiotic relationship known as mycorrhizae with their host plants. This symbiosis typically exhibits reciprocity via a mutual exchange of nutrients, particularly P, N, and zinc (Zn), between fungi, plants, and soil. The symbiotic relationship between plants and microorganisms has the potential to enhance the capacity of host plants to acquire essential mineral nutrients, including nitrogen, phosphorus, sulfur, zinc, and copper. Additionally, this symbiotic association has been observed to confer improved resistance to various forms of stress in host plants [44]. Research on the symbiotic association between AMF and plants has shown various effects of AMF on plant productivity, species diversity, and ecological stability [45]. Plant community diversity and ecosystem production were found to be influenced by AMF [46]. Carbon transport in *Robinia pseudoacacia* to the rhizosphere soil and in the plant body of *Amorphophallus konjac* can be promoted by AMF colonization, and the colonization of AMF and the absorption of P by host crops can also be promoted by intercropping cultivation, thus increasing plant biomass [47].

Furthermore, the application of AMF not only resulted in an increase in the levels of *Glomus*-related proteins and macroaggregates in the soil but also had a significant effect on reducing the concentration of cadmium (Cd) in both the soil and loam [48]. The species composition of microorganisms varies across different regions, leading to distinct variations in the diversity and structure of AMF communities. These variations can be attributed to differences in dominant plant species, climatic conditions, and soil parent materials.

In the present study, *Glomus* was found to be the dominant AMF in northwestern Yunnan and a prevalent fungus found in the rhizosphere soil of plants. This genus was

observed to promote plant development and reduce disease occurrence under organic field conditions [49]. Additionally, *Acaulospora*, one of the most vital genera of AMF belonging to the family Acaulosporaceae, was found to increase soil P availability by enhancing the secretion of acid phosphatase in mycorrhizal roots [50].

## 5. Conclusions

The largest AMF community abundance was found in shrubland, while woodland exhibited the lowest. *Glomus* demonstrated to be the most abundant genus, followed by *Paraglomus*. There was a significant association between the presence of *Glomus* and AN in the soil. Additionally, a significant and positive correlation was observed between *Glomus* and N in grass shoots; *Paraglomus* was positively correlated with the content of AN during the summer season. Similarly, *Glomus* displayed a significant and positive correlation with TP in soil, as well as with P in grass roots during the winter season. To ensure the stability and favorable progression of the subalpine ecosystem in northwestern Yunnan, preserving the existing vegetation in this region and maintaining the balance and stability of the AMF community are of utmost significance. Successful grass preservation would ensure the conservation of essential nutrients in the soil, thus establishing a resilient basis for the restoration of biodiversity and the overall ecosystem balance in subalpine regions.

**Author Contributions:** Conceptualization, F.Z.; methodology, J.Y.; software, J.Y.; validation, F.Z. and Y.Z.; formal analysis, J.G. and H.D.; investigation, J.G. and H.D.; resources, H.D.; data curation, J.G.; writing—original draft preparation, W.L.; writing—review and editing, W.L.; visualization, Y.B.; supervision, Y.H.; project administration, Y.H.; funding acquisition, J.Y. All authors have read and agreed to the published version of the manuscript.

**Funding:** This research was funded by the Geological Survey Project of China Geological Survey (grant number DD20230482), the University-enterprise Cooperation Project (grant number 20200618), and the Fund Project of Education Department of Yunnan Province (grant number 2021J123).

**Data Availability Statement:** Data are contained within the article.

**Conflicts of Interest:** Author Hengwen Dong is employed by the company Kunming Prospecting Design Institute of China Nonferrous Metals Industry Co., Ltd. The remaining authors declare that the research was conducted in the absence of any commercial or financial relationships that could be construed as a potential conflict of interest. The Kunming Prospecting Design Institute of China Nonferrous Metals Industry Co., Ltd. had no role in the design of the study; in the collection, analyses, or interpretation of data; in the writing of the manuscript, or in the decision to publish the results.

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
