# Peer review of "Characteristics of the Rhizospheric AMF Community and Nutrient Contents of the Dominant Grasses in Four Microhabitats of the Subalpine Zone in Northwestern Yunnan, China"

_forests, doi:10.3390/f15040656_

Round 1

Reviewer 1 Report

Comments and Suggestions for Authors

The issues discussed are essentially related to the importance of biodiversity in the functioning of ecosystems, especially in subalpine regions. The results obtained by the Authors may not only expand knowledge regarding the functioning of subalpine regions, but may also contribute to the effective protection of these regions.

Line 88 - samples were taken in four microhabitats - it would be worth providing the number and size of the plots from which the samples were taken. Information on the total number of samples taken would be valuable.

Line 187, Table 1 - although the abbreviations used in the table have already been explained in the methods section, it does not hurt to repeat them in the table title; the same in the case of Table 2 (line 196); the same in the case of Figure 9 (line 271) and Figure 10 (line 287)

The Conclusions refer only to the genus Glomus. It was previously found that the genus Paraglomus was also abundant. Maybe it's worth mentioning it too. It would also be worth referring to other AMF, at least in one sentence. In this way, the importance of the genus Glomus for the studied ecosystems would become even more important.

Reviewer 2 Report

Comments and Suggestions for Authors

The main handicap of this work is the Materials and Methods section. The presentation and description of methodologies need to be improved.

Abstract

Line 22: “… in the other 11 grass species” or “… in the other 10 grass species”

Material and Methods

Is the climate characterization of the study area or Yunnan Province? Clarify

What criteria were used in the selection of microhabitats? You should mention in the text the four microhabitats studied and do a brief characterization of them.

In "2.2 Sampling of soil and grasses" you just mention how you collected the soil samples from the rhizosphere. It remains to be mentioned the procedures used in harvesting vegetation. Were vegetation and soil samples taken from the same points? How many points did you sample (repetitions)? What criteria were used in selecting the points? Were grass species sampled separately in each microhabitat?

Lines 94-97: You only used soil brushed from the roots for analysis, right? If only the soil attached to the roots was not used, the thickness of soil around the roots used in the analysis should be mentioned. This information should be clear in the text

Lines 105-106: What are the sieve mesh units?

Line 144: “…including soil physicochemical properties…”  Were the physical soil properties also analyzed? In the methodology, you only refer to the determination of chemical soil properties! This information needs to be corrected.

Lines 148-155: This information is not data processing, but rather data collection. Please place it in the Material and Methods section.

Results

Were the grasses analyzed separately? This must be described in Material and Methods.

“Figure 2. Soil nutrient contents of grasses in four microhabitats”.

Are the data presented in Figure 2, 3 from soil or plant biomass? The title of the item "Nutrient contents in grass" leads me to think that it is from biomass, but the caption of the Figures says "Soil". Please, clarify.

You should improve the presentation of Table 1 to make it easier to read.

All Figures and Tables should be read independently. Therefore, you must include in the Title of Table 1 and 2 the meaning of the acronyms used for the variables.

Do the values presented in Table 1 and 2 refer to the “mean ± standard deviation”? Put this information in the title of the Table.

“3.2 Soil physicochemical properties” or “3.2 Soil chemical properties”

3.6 Correlation between soil physicochemical properties and AMF communities

Discussion

Good

4.1 Characteristics of soil physicochemical properties and grass nutrients

4.3 Effects of AMF communities on soil physicochemical properties

Reviewer 3 Report

Comments and Suggestions for Authors

The authors characterized the abuscular mycorrhizal community and analyzed the true contents of predominant grasses in several microhabitats. I believe that the study is relevant to the aims and scope of forests. However, I would like to reject the publication of the paper as it is apparent that the researchers collected only two replications in each treatment (Line 16). Although it was mentioned that they had three replications for molecular analysis of soil microorganisms, it is still considered a subsample (Line 131). I am happy to further evaluate the manuscript to see if this problem will be corrected.

Comments on the Quality of English Language

The manuscript requires very minor English editing.

Round 2

Reviewer 2 Report

Comments and Suggestions for Authors

I consider that the article meets the conditions to be published.

Reviewer 3 Report

Comments and Suggestions for Authors

Manuscript ID: forests- 2897958

The authors characterized the abuscular mycorrhizal community and analyzed the nutrient contents of predominant grasses in several microhabitats. I believe that the study is relevant to the aims and scope of forests.

Comments

  • The manuscript must be reviewed for grammatical (e.g. a influence in line 292) and numerous typographical errors, such as the failure to have a space in between words in the caption of Figures 7-10 as well as inconsistency in writing the titles in references.
  • In Lines 89 -108, numbers 1 to 9 should be written in full (e.g. one, two, three,.. nine, 10) except for numbers and unit combinations (e.g. 2 cm).
  • Numbers at the start of the sentence should be written in full (e.g. see Lines 96, 103).
  • The authors must be consistent in writing number and unit if there is no space or with space (e.g. 2cm vs 2 cm).
  • The authors must also check the consistency of the references.

  1. In reference 33 (Lines 516-517), [J] should be writtend after the last word of the title, which is after diversity.
  2. Check the consistency of writing titles, e.g. Lines 470, 477, 486, 492,505, and 512. Please check all refences and not just the one mentioned here.
Comments on the Quality of English Language
  • The manuscript must be reviewed for grammatical (e.g. a influence in line 292) and numerous typographical errors, such as the failure to have a space in between words in the caption of Figures 7-10 as well as inconsistency in writing the titles in references.
